# The Impact of Dose Rate on DNA Double-Strand Break Formation and Repair in Human Lymphocytes Exposed to Fast Neutron Irradiation

**DOI:** 10.3390/ijms20215350

**Published:** 2019-10-28

**Authors:** Shankari Nair, Monique Engelbrecht, Xanthene Miles, Roya Ndimba, Randall Fisher, Peter du Plessis, Julie Bolcaen, Jaime Nieto-Camero, Evan de Kock, Charlot Vandevoorde

**Affiliations:** 1Department of Radiochemistry, South African Nuclear Energy Corporation, Pretoria 001, South Africa; 2Radiobiology, Radiation Biophysics Division, Department of Nuclear Medicine, iThemba LABS, Cape Town 7131, South Africa; mengelbrecht002@gmail.com (M.E.); xmuller@tlabs.ac.za (X.M.); rminnis@tlabs.ac.za (R.N.); rfisher@tlabs.ac.za (R.F.); pdp@tlabs.ac.za (P.d.P.); jbolcaen@tlabs.ac.za (J.B.); jaime@tlabs.ac.za (J.N.-C.); evan@tlabs.ac.za (E.d.K.); 3Department of Medical Biosciences, University of the Western Cape, Cape Town 7535, South Africa

**Keywords:** DNA double-strand breaks, γ-H2AX, neutrons, low dose rate, high dose rate, lymphocytes

## Abstract

The lack of information on how biological systems respond to low-dose and low dose-rate exposures makes it difficult to accurately assess the carcinogenic risks. This is of critical importance to space radiation, which remains a serious concern for long-term manned space exploration. In this study, the γ-H2AX foci assay was used to follow DNA double-strand break (DSB) induction and repair following exposure to neutron irradiation, which is produced as secondary radiation in the space environment. Human lymphocytes were exposed to high dose-rate (HDR: 0.400 Gy/min) and low dose-rate (LDR: 0.015 Gy/min) *p*(66)/Be(40) neutrons. DNA DSB induction was investigated 30 min post exposure to neutron doses ranging from 0.125 to 2 Gy. Repair kinetics was studied at different time points after a 1 Gy neutron dose. Our results indicated that γ-H2AX foci formation was 40% higher at HDR exposure compared to LDR exposure. The maximum γ-H2AX foci levels decreased gradually to 1.65 ± 0.64 foci/cell (LDR) and 1.29 ± 0.45 (HDR) at 24 h postirradiation, remaining significantly higher than background levels. This illustrates a significant effect of dose rate on neutron-induced DNA damage. While no significant difference was observed in residual DNA damage after 24 h, the DSB repair half-life of LDR exposure was slower than that of HDR exposure. The results give a first indication that the dose rate should be taken into account for cancer risk estimations related to neutrons.

## 1. Introduction

The integrity of an individual’s genome is continuously challenged by both endogenous and environmental mutagens. These damaging agents can induce a wide variety of lesions in the DNA, such as single-strand breaks (SSBs), double-strand breaks (DSBs), oxidative lesions and pyrimidine dimers [1]. Ionising radiation (IR) was one of the first environmental agents identified as being mutagenic by inducing various types of DNA damage, of which DNA DSBs are considered to be the most deleterious [2]. Inaccurate repair or lack of repair of DSB can lead to cell death or mutations. The latter has been confirmed by experimental evidence on a causal link between the generation of DSBs and the induction of chromosomal translocations with tumourigenic potential [3]. To maintain the genomic integrity of DNA, cells have a complex set of signalling pathways to repair these DSBs, known as the DNA damage response pathway. The two major DSB repair pathways in mammalian cells are nonhomologous end joining (NHEJ) and homologous recombination (HR) pathways [4]. The HR pathway is error-free but requires an intact homologous template, while the error-prone NHEJ is the most prominent pathway for DSB repair in mammalian cells and does not require a homologous sequence to guide the repair [5]. The number of induced DNA DSBs is proportional to the radiation dose, whereas the complexity of the DSB and the repair ability of the DNA damage response machinery of a specific cell can be influenced by the radiation quality [6]. In addition, recent evidence shows that different particles with similar linear energy transfer (LET) values induce DSB damage of different complexities, pointing to a critical impact of the particle track core diameter on the type of DNA damage [7].

The induction of a DSB results in the activation of a phosphatidylinositol-3-OH-kinase-like kinase, ataxia telangiectasia mutated protein (ATM). A critical target of ATM is the phosphorylation of the C-terminus of the histone variant H2AX, known as γ-H2AX, which is one of the earliest steps following DNA DSB induction [8]. This process rapidly spreads over an extensive region surrounding the DSB, leading to the formation of a γ-H2AX focus. There is a close correlation between the number of DNA DSBs and γ-H2AX foci and between the rate of foci loss and DSB repair. The latter makes it an appropriate assay to monitor DSB formation and repair in individual cells after IR exposure [9,10,11]. The demonstration of precise γ-H2AX localisation at the sites of DNA DSBs was achieved by means of a laser scissor experiment, where DSBs were introduced through a pulsed laser microbeam [12]. Subsequent immunocytochemistry showed that γ-H2AX forms precisely along this track of DSBs [13]. It is also a very sensitive assay, since the formation of γ-H2AX foci has been detected in fibroblasts and lymphocytes following radiation doses as low as 1 mGy, and the increase in foci yields is strongly dose-dependent [14]. Over the past 20 years, the γ-H2AX foci assay has received growing attention and is now well established as a biomarker to detect radiation-induced DNA DSBs [15]. Furthermore, several studies have shown that increased γ-H2AX foci levels demonstrate genomic instability, known to be a hallmark of cancer development [16].

The mechanisms of DNA DSB induction and repair after an acute and short-term exposure to IR have been well studied [17]. However, significant uncertainties exist on the effects of low doses and low dose rates of IR, which remains the subject of active debate [18,19]. Over the past decades, the validity of the linear no-threshold (LNT) hypothesis and its dose and dose-rate independence with respect to risk has been questioned by several regulatory bodies [20,21]. Unfortunately, reports that describe the induction and repair of DNA DSB after chronic low-dose and low dose-rate exposure to IR in mammalian cells are limited. While this remains a topic of active debate, most regulatory bodies use a dose and dose-rate effectiveness factor (DDREF) to reduce the cancer risk estimates from high-dose and high dose-rate studies down to lower doses and low dose rates. Despite the observed sparing effect of chronic low dose-rate exposures, it still remains unclear if the kinetics and accuracy of DSB repair depend on the dose rate. Therefore, research on the dose and dose-rate dependence of cancer risk has been listed as one of the key research topics in the recently published strategic research agenda of the Multidisciplinary European Low Dose Initiative (MELODI) [22]. The consortium encourages research to improve understanding of the basic mechanisms underlying the observed dose and dose-rate effects, specifically for internal exposures, inhomogeneous exposures and different radiation qualities. In addition, while it has long been considered that the dose-rate dependence of high-linear energy transfer (LET) radiation qualities is negligible, such as for neutrons and alpha-particles, there is a growing body of evidence showing an inverse dose-rate effect for carcinogenesis induced by high LET radiation, with an increasing effect observed with decreasing dose rate [23,24,25,26].

Low dose-rate effects are generally a characteristic of environmental (high radon exposures in buildings) and occupational (aircrew on long-distance flights or naturally occurring radioactive materials in the oil extraction industry) exposures and are also applicable to the space radiation environment, where astronauts are spending more and more time [22,27]. While there is a number of studies on DNA DSB formation after low dose-rate versus high dose-rate photon irradiation [28,29,30,31,32,33], there is currently limited information available on the dose-rate effects of high-linear energy transfer radiation qualities on DNA DSB formation and repair [34,35]. Due to an incomplete understanding of how biological systems respond to the low doses and low dose rates of cosmic radiation (including protons, helium nuclei, high-Z high-energy particles and secondary neutrons) and the limited human epidemiological data for these radiation qualities, it is currently difficult to accurately estimate the risk of carcinogenesis and other health effects due to long-term chronic IR exposure of astronauts in space. For this reason, space radiation remains the number one risk to astronaut health, and the existing uncertainties limit the planning of manned interplanetary missions [36]. Particularly for chronic exposure to neutrons, which are produced as secondary radiation during space missions as a result of nuclear interactions with the spacecraft walls and the human body, there is currently limited data available on the associated cancer risks [37,38]. The Radiation Assessment Detector (RAD) on-board the Mars Science Laboratory (MSL) rover Curiosity provided the first radiation measurements on the surface of Mars in order to fully characterise the Martian radiation environment. This included the measurements of energetic charged and neutral particles and the radiation dose rate on the surface of Mars since the landing of the rover in 2012. The neutron spectrum at the surface consists of neutrons with a very broad energy range coming from above and comparatively low-energy neutrons coming from below. The measured dose rate for the Martian neutron spectrum ranging from 8 to 740 MeV was 0.014 ± 0.004 mGy/day [39,40]. An additional uncertainty is the weighting factor (w_R_) for neutrons, which is used to convert the physical absorbed dose (Gy) into an equivalent dose (Sv), due to limited biological data for high neutron energies on relevant biological endpoints and tissue types [41]. Next to the health effects for astronauts during deep space missions, the long-term cancers risks related to secondary neutron production during radiotherapy remain a controversial topic, particularly when high-energy X-rays (E > 10 MeV) and protons are used [42].

To this end, the induction and repair of DSBs was investigated in vitro in whole blood of human adult volunteers after exposure to a fast neutron beam at two different dose rates: low dose rate (0.015 Gy/min) and high dose rate (0.400 Gy/min). It was decided that DNA DSBs in isolated peripheral blood lymphocytes would be evaluated in this study, since they are widely used for the detection and evaluation of radiation-induced DNA damage in humans. The hematopoietic system contains some of the most radiosensitive and easily sampled cells in the human body. Furthermore, circulating lymphocytes are a synchronic noncycling population with a low percentage of nucleosomal H2AX variant, resulting in low cell cycle effects and low γ-H2AX background levels [43].

## 2. Results

### 2.1. Dose Response Curves for γ-H2AX Foci Induction after LDR Versus HDR Neutron Irradiation

The γ-H2AX foci yield after exposure to *p*(66)/Be(40) neutrons at a dose rate of 0.400 Gy/min (HDR) and 0.015 Gy/min (LDR) are presented in Table 1 and Figure 1, with representative images presented in Figure 2. The gradual increase in the number of γ-H2AX foci with increasing dose was statistically significant (*p* < 0.05). It was noted that the γ-H2AX foci formation after exposure to HDR was higher than that after LDR, which appeared to be a statistically significant dose-rate effect (*p* < 0.05). This was the result of simultaneous DNA DSB induction and repair during LDR exposure, resulting in a lower number of γ-H2AX foci by the completion of exposure. Regression analysis indicated that the mean number of DNA DSBs per cell after exposure to graded doses of either dose rates was best fitted to a second-order polynomial equation. There was a sharp increase in the number of neutron-induced DNA DSBs at lower doses, and it appeared to reach a plateau at 1 and 2 Gy. A potential explanation for this observation is that the actual number of foci may have been higher than the observed number. It is anticipated that more DNA DSBs will be induced in close proximity to each other at higher neutron doses, which might be joined into a single larger focus that is only scored as one DNA DSB. Another option is that cells with a high level of DNA damage went into apoptosis and could not be detected by the MetaCyte scoring system due to morphological changes; hence, these cells were not considered in the final average number of γ-H2AX foci per cell.

A ratio was calculated of the HDR over the LDR induced DNA DSB values, ranging between 1.16 and 1.87 for neutron doses of 0.125 to 2.000 Gy, with a maximum difference between LDR and HDR foci yields observed at 0.250 Gy (Table 2). On average, HDR neutron irradiation induced 40% more DNA DSBs per cell compared to the mean number of DNA DSBs per cell induced at LDR. 

### 2.2. Repair Kinetics of γ-H2AX foci after LDR Versus HDR Neutron Irradiation

The average number of DNA DSB was investigated in isolated lymphocytes of four donors at different time points (0.5, 2, 4, 8, 12 and 24 h) postirradiation to 1 Gy of HDR (0.400 Gy/min) and LDR (0.015 Gy/min) neutrons (Table 3, Figure 3). Statistically, there was no overall difference in the number of DNA DSBs at the two dose rates over a 24 h period. The maximum number of foci per cell was reached at 2 h postirradiation. This was seen for both the HDR (mean value 5.75 ± 0.05) and LDR (mean value 5.60 ± 0.48) exposures. After the maximum peak, the number of foci decreased gradually for both dose rates, to 1.65 ± 0.64 foci/cell (LDR) and 1.29 ± 0.45 (HDR) at 24 h postirradiation. The absolute number of foci per cell at 24 h was significantly different from the background level of γ-H2AX foci in nonirradiated cells (*p* < 0.005); hence, the number of residual DNA DSB did not return to background levels after 24 h postirradiation for both dose rates. Twelve hours postirradiation, the foci frequencies for samples irradiated at LDR exceeded those of the HDR samples (Figure 4). This observation was still present 24 h postirradiation, but the difference was not statistically significant at both time points.

To better compare the results of different dose rates, the foci numbers were normalised to the maximum number observed at 2 h postirradiation (Figure 4). Foci induced by LDR neutrons disappeared slower than those induced by HDR neutrons at the latest time points. When the residual foci number at 24 h postirradiation was subtracted from the foci yields obtained at earlier time points postirradiation, all remaining foci data of the present study for 0.400 Gy/min were consistent with a repair half-life of 8.6 h. For dose rate of 0.015 Gy/min, the repair half-life was longer, namely, 12 h. Notwithstanding, the significant difference in initial DNA DSB induction between HDR and LDR neutron exposure, the half-life of foci disappearance was marginally longer for LDR neutrons than that for HDR neutrons.

## 3. Discussion

The understanding of the mechanisms underlying the biological effects observed for low dose-rate exposure to fast neutrons are very important in the field of radiation protection for cancer risk estimations related to long-term space flights and commercial transatlantic airplane flights. In addition, a better understanding of the radiobiological effects of low neutron doses is relevant for radiotherapy patients, where neutrons can form a significant part of the stray radiation produced during treatment. Our results showed clear differences in the induction and repair after LDR versus HDR *p*(66)/Be(40) neutron irradiation of peripheral blood lymphocytes. Interestingly, the difference in DNA DSB induction was particularly high at lower doses, as illustrated in Table 1. However, the effect was not so pronounced for the lowest dose of 0.125 Gy, possibly because the exposure to the lowest dose rate of 0.015 Gy/min for this radiation dose lasted only 8 min. These results provide a better understanding of the health effects related to low dose-rate neutron radiation, particularly at low doses, which are arguably the most common type of human exposure to neutrons [29].

For the estimation of stochastic effects, such as solid cancers and leukaemia after external radiation exposure, most of our knowledge is based on the Life Span Studies (LSS) from the Japanese atomic bomb survivors, who were exposed to high dose rates. In order to quantify radiation risks at exposure scenarios relevant for radiation protection, one has to extrapolate this data obtained at high doses and high dose rates down to low doses and low dose rates [44]. However, this extrapolation comes with large uncertainties, and the LNT theory remains a topic of active debate. Epidemiological studies on low-dose effects remain challenging, and a growing amount of evidence suggests that there is no scientific proof for low-dose and low dose-rate effects [45,46]. The current LNT approach might have a negative impact on public perception, policy statements and future directions for the radiation protection framework. Regulatory bodies have adopted the DDREF for low-LET radiation, which implies that radiation delivered at low total doses or low dose rates are less effective for cancer induction, but not for high-LET radiation [47]. However, a number of sources have proposed a different, potentially more correct approach to separately consider a ‘low-dose effectiveness factor’ (LDEF) and a ‘dose-rate effectiveness factor’ (DREF) for risk estimate calculations [48]. These calculations are important for populations living in contaminated areas after nuclear accidents and high natural background radiation areas as well as individuals who are occupationally exposed to ionizing radiation, such as aircrew and astronauts. Although there is not a precise definition, one can generally consider a low dose to be defined as <100 mGy and a low dose-rate as <0.1 mGy/min, as stated by the United Nations Scientific Committee on the Effects of Atomic Radiation (UNSCEAR) [49]. In the present study, we were limited by the detection limits of the electronics of the clinical neutron therapy beam line; therefore, a dose rate of 0.015 Gy/min was the lowest that could be achieved. However, an approximately 25 times reduction of dose rate in our study could already illustrate a dose-rate effect for fast neutrons on DNA DSB induction evaluated 30 min postirradiation.

While the LSS cohort received an acute exposure mainly to high-energy photons, with a small contribution from neutrons, the current DDREF only applies to low-LET radiation. Effects of high-LET radiation were reported to have a small or no dose-rate dependence in contrast to low-LET radiation [50]. In order to evaluate the risks from fission neutron exposures (average energy of approximately 2 MeV), it was therefore previously decided that the relative biological effectiveness (RBE) values from higher doses and dose rates would be used directly, thereby avoiding the use of low dose-rate data for γ-rays and its associated uncertainties [51]. However, DDREF remains, together with the radiation quality factor function, one of the largest uncertainties for radiation risk estimates related to high-LET radiation [50]. Most existing neutron RBE and dose-rate results are based primarily on experiments with exposures to neutron energies below 10 MeV, while simulation and dosimetry studies illustrate that neutron energies in the space environment go well beyond 10 MeV, up to even GeV [52]. These high-energy neutrons pose a challenge not only for dosimetry and monitoring but also for radiobiologists, since the biological effects of high-energy neutrons are hardly studied.

The DNA repair protein foci assays (e.g., γ-H2AX, 53BP1, pATM) are considered to be among the most sensitive endpoints currently in use to study low-dose and low dose-rate effects for in vitro scenarios [53]. For all dose points, the LDR neutrons induced a reduced number of foci per cell compared to the HDR. As previously mentioned, the actual number of DNA DSBs per cell produced by 1–2 Gy neutron doses may have been greater than the observed number of foci because closely located DNA DSBs may have been joined into a single focus. Unfortunately, the microscope system that was used for this study could not provide 3D information on the γ-H2AX foci size, for which confocal microscopy would be required. The average spot size (2D) that could be retrieved from the MetaCyte software for the different experimental conditions did not provide consistent results (data not shown here). The results indicate that the γ-H2AX foci levels saturate at the higher neutron doses and supports the idea that clustering of DNA DSBs occurs [54].

To quantify the absolute number of DNA DSB per unit of radiation dose induced by LDR versus HDR exposure, the K coefficient (K, %) was calculated, as previously proposed by Ulyanenko et al. [29]. The K coefficient for the HDR exposure was higher than that for the LDR exposure for all doses, which is in agreement with the previous study [29]. The maximum difference of 1.87 was observed between the K coefficients of the two dose rates at 0.250 Gy, which is also clear from the ratios presented in Table 2. At a similar absorbed dose of 0.240 Gy ^137^Cs γ-radiation, Ulyanenko et al. reported a difference of 2.43 between the low dose rate (0.0001 Gy/min) versus the high dose rate (0.030 Gy/min) [29]. Moreover, our results at 30 min postexposure agree with the data reported by Kotenko et al. on Chinese hamster V79 cells, where the exposure to Co^60^ γ-rays at a lower dose rate (0.001 Gy/min) led to a reduced number of γ-H2AX foci per cell compared to the HDR (0.4 Gy/min) exposure [28].

Further differences in γ-H2AX foci formation between the HDR and LDR neutrons were noted when analysing repair kinetic data. In the repair kinetic experiments on human lymphocytes, the number of neutron-induced γ-H2AX foci did not seem to revert back to the background levels 24 h after exposure to 1 Gy of HDR (0.400 Gy/min) and LDR (0.015 Gy/min), suggesting that it takes longer to reach background levels when using neutrons compared to low-LET Co^60^ γ-ray results reported by other groups [55]. Moreover, the remaining DNA DSBs showed a half-life of 12 h after a dose rate of 0.015 Gy/min and a shorter half-life of 8.6 h after a dose rate of 0.400 Gy/min. The maximum number of γ-H2AX foci per cell was reached at 2 h postirradiation. This was observed for both dose rates, which is counterintuitive as one would expect the maximum number of γ-H2AX foci at earlier time points postirradiation. At 12 and 24 h postirradiation, the number of foci was higher after LDR neutrons compared to HDR neutrons; however, this difference was not significant. The difference in repair half-life are in agreement with previous observation for low-LET (γ-rays) radiation, where the LDR resulted in a longer pATM repair half-life compared to the HDR in human mesenchymal stem cells [29]. Studies have shown that a slower disappearance of DNA DSBs has been associated with an increase in miss-joining events and thus the formation of chromosome aberrations (e.g., dicentrics) as there is more time for exchange between the ends of the free DNA DSB [56,57]. This is in line with studies that have shown an increased induction of mutation or chromosomal aberrations and enhancement of cell-killing at lower dose rates [26,58,59]. A recent study provided more insight in this inverted dose-rate effects for low-LET radiation by investigating dose rate-induced changes in cell-cycle distribution. This showed that the change in S phase fraction during irradiation might be responsible for some of the inverse dose-rate effects [60]. Work by Turner et al. on peripheral mouse lymphocytes after prolonged in vivo exposure to LDR (0.0031 mGy/min) and HDR (1.03 Gy/min) X-rays showed that the residual γ-H2AX foci levels after 24 h were significantly higher in the lymphocytes exposed to HDR X-rays compared to the LDR exposure at 1.1 and 2.2 Gy total body doses [32]. Our results are supported Ishizaki et al. where HDR ^137^Cs γ-ray exposure induced DNA DSB in a dose-dependent manner and the LDR (0.0003 Gy/min) exposure induced almost no foci, even at the highest dose point of 5 Gy, compared to the HDR (1.8 Gy/min) at 3 h postirradiation [30]. The observed difference between HDR and LDR was much larger in the latter study; however, the LDR was 15 times lower than the LDR used in our study. To the best of our knowledge, only one study has investigated the γ-H2AX foci formation after HDR (0.1 Gy/min) and LDR (0.003 Gy/min) 24 h after exposure to a ‘medium-energy’ neutron field (^9^Be + d reaction, with a deuteron energy of 13 MeV) with a tissue kerma-averaged mean neutron energy of 5.8 MeV. This study was performed with human mammary epithelial cells (MCF10A) and did not show a significant difference in residual DNA DSB at 24 h after a neutron dose of 1 Gy, which is in line with our results on DNA repair kinetics [35].

In the current study, a low dose-rate (0.015 Gy/min) fast neutron beam with a fluence-weighted average energy of approximately 29.8 MeV was used. This dose rate is still high in comparison to the dose rates in the space radiation environment, aircraft altitudes or areas with high-LET natural background radiation (radon-prone areas) [61]. However, the LDR used here was the minimal dose rate that could be obtained for the *p*(66)/Be(40) neutron beam line. 

In summary, this study showed a significant difference in the induction of γ-H2AX foci after in vitro exposure of peripheral blood lymphocytes to fast neutrons at LDR (0.015 Gy/min) versus HDR (0.400 Gy/min). The number of γ-H2AX foci gradually decrease over a period of 24 h, with the γ-H2AX foci induced by low dose rate (0.015 Gy/min) decreasing at a slower rate compared to the high dose rate (0.400 Gy/min). These results are a first indication that the dose rate has an effect on the efficiency of DNA DSB induction and repair for fast neutrons, which contributes to the discussion of whether LDEF or DREF should be recommended for high-LET radiation. However, additional studies are required on the outcome of DNA DSB repair for fast neutrons at different dose rates in order to make a stronger conclusion related to cancer risks. These results will assist in the estimation of the long-term carcinogenic risks due to chronic exposure to secondary high-energy neutrons relevant to the space radiation environment and radiation protection for high-LET radiation in general.

## 4. Materials and Methods

### 4.1. Sample Collection

Blood samples were collected by venepuncture from four healthy adult volunteers aged between 26 and 60 (one male and three females), from whom informed consent was obtained prior to the experiments. The experiments were approved by the Biomedical Research Ethics Committee (BMREC) of the University of the Western Cape (reference number: 15/4/100; original approval date: 19 August 2015). Blood samples were collected in lithium–heparin collection tubes and diluted (1:1) in Roswell Park Memorial Institute (RPMI) medium (Lonza, Walkersville, MD, USA), supplemented with 10% fetal calf serum (Gibco, Dun Laoghaire, Dublin, Ireland) and 1% penicillin and streptomycin (Lonza, Walkersville, MD, USA) in sterile 2.0 mL cryogenic vials (NEST Biotechnology Co., Ltd., Wuxi, Jiangsu, China). Cells were incubated in a humidified 5% CO_2_ incubator at 37 °C before irradiation.

### 4.2. Irradiation Experiments

Each blood sample was exposed to a fast neutron beam, which was produced by the reaction of a 66 MeV proton beam on a Beryllium target (p + ^9^Be→*n* + ^9^B-1.85 MeV, plus several breakup reactions), resulting in a neutron spectrum with a fluence-weighted average energy of approximately 29.8 MeV for the 29 × 29 cm field used [62]. The neutron beam was modified by 0.8 cm thick iron flattening filters and a 2.5 cm thick polyethylene hardening filter. The irradiations were carried out in a horizontal direction (gantry angle of 270°) in a 37 °C water tank. Each exposure condition was carried out at two different dose rates: LDR of 0.015 Gy/min and HDR of 0.400 Gy/min. The output factor (1.097 Gy/MU) was determined using a 0.5 cm^3^ tissue equivalent ionisation chamber at the same position of the blood samples (depth in water tank: 5.2 mm). For the dose response experiments, the blood samples were exposed to radiation doses of 0.125, 0.250, 0.500, 1.000 and 2.000 Gy. These samples were kept at 37 °C for 30 min, followed by arrest on ice. For the repair kinetic experiments, cultures were irradiated with a dose of 1.000 Gy and incubated at 37 °C for different time points postirradiation (0.5, 2, 4, 8, 12, 24 h) to allow foci formation and repair. Nonirradiated control samples were included for each experimental condition and incubated for the same time.

### 4.3. Lymphocyte Isolation and Immunofluorescence γ-H2AX Staining

After the incubation, lymphocytes were isolated from whole blood using a density gradient cell separation medium (Histopaque-1077, Merck, Modderfontein, Johannesburg, South Africa) at a 2 (diluted blood) over 1 (gradient medium) ratio. The layered solution was centrifuged at 2130 rpms for 20 min. Afterwards, the interface between plasma and density medium was carefully transferred to a new tube and washed twice with phosphate-buffered saline (PBS) at 1500 rpms for 10 min. Cells were counted, and a lymphocyte suspension of approximately 120,000 cells/0.25 mL was centrifuged onto coated slides (X-tra adhesive slides, Leica Biosystems, Buffalo Grove, IL, USA) using a cytospin (Cellspin I, Tharmac^®^ GmbH). Three slides were prepared for each exposure condition. The slides were fixed in PBS containing 3% freshly prepared paraformaldehyde (PFA) for 20 min, followed by storage overnight in PBS containing 0.5% PFA. Immunostaining was performed as described in [63]. Finally, slides were scanned automatically using the MetaCyte software module of the Metafer 4 scanning system with a 40× objective and a z-stack of 10 focal planes, as described in [9]. For each slide, approximately 1000 lymphocytes were captured, and the average number of γ-H2AX foci per scanned slide was derived from the MetaCyte software. For every experimental condition, at least three different slides were scanned, resulting in more than 3000 analysed cells per exposure condition. The number of γ-H2AX foci induced by the neutron irradiation was obtained by subtracting the mean number of γ-H2AX foci in the nonirradiated controls from the mean γ-H2AX foci number scored in the irradiated samples.

### 4.4. Statistical Analysis

Statistical analysis and curve fitting was performed using Microsoft Office Excel 2013 (Microsoft Corporation, Washington, DC, USA) and GraphPad Prism Software Version 5.00 for Windows (GraphPad Software, San Diego, CA, USA). Data represents the mean number of γ-H2AX foci ± standard deviation of two independent experiments on two different days. All data represent the mean values obtained for 4 different donors. The *p*-value is defined as the probability that the total sum of squares for the random dataset would be greater than that for the observed one.

## Figures and Tables

**Figure 1 ijms-20-05350-f001:**
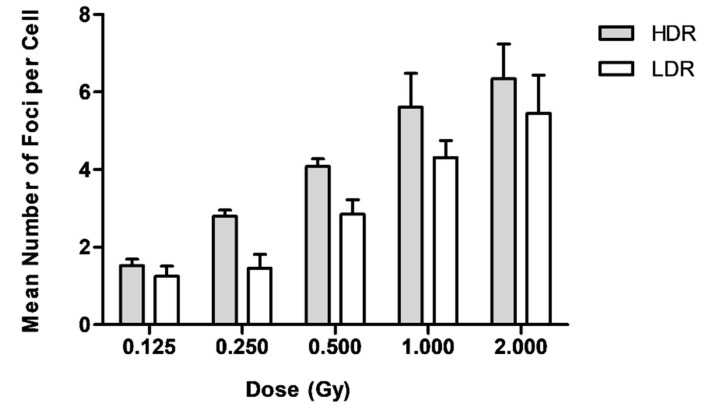
The number of γ-H2AX foci as a function of dose for lymphocytes exposed to HDR (0.400 Gy/min) and LDR (0.015 Gy/min) neutrons. Data represents the mean number of γ-H2AX foci ± standard deviation of four healthy volunteers over two independent experiments.

**Figure 2 ijms-20-05350-f002:**
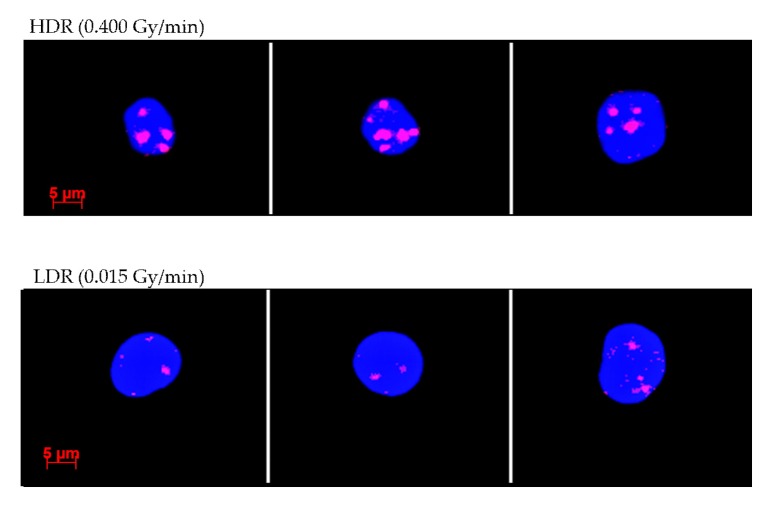
Images retrieved from the Metafer automatic image analysis software representing γ-H2AX foci in lymphocytes exposed to HDR (0.400 Gy/min—top) and LDR (0.015 Gy/min—bottom) neutrons. Red dots indicate γ-H2AX foci, while the nuclei are stained blue with a fluorescent DNA stain, namely 4′,6-diamidino-2-phenylindole (DAPI).

**Figure 3 ijms-20-05350-f003:**
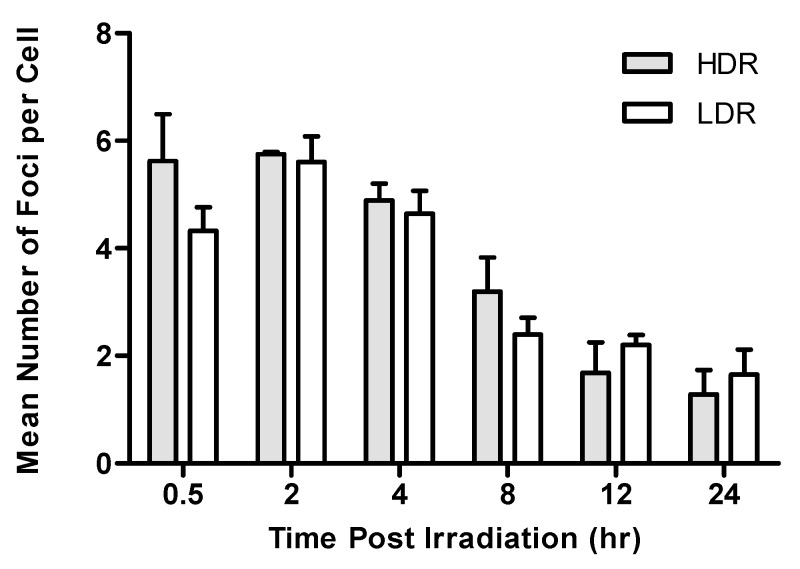
The number of γ-H2AX foci as a function of dose for human lymphocytes exposed to 1 Gy of HDR (0.400 Gy/min) and LDR (0.015 Gy/min) neutrons. Data represents the mean number of γ-H2AX foci ± standard deviation of four healthy volunteers over two independent experiments.

**Figure 4 ijms-20-05350-f004:**
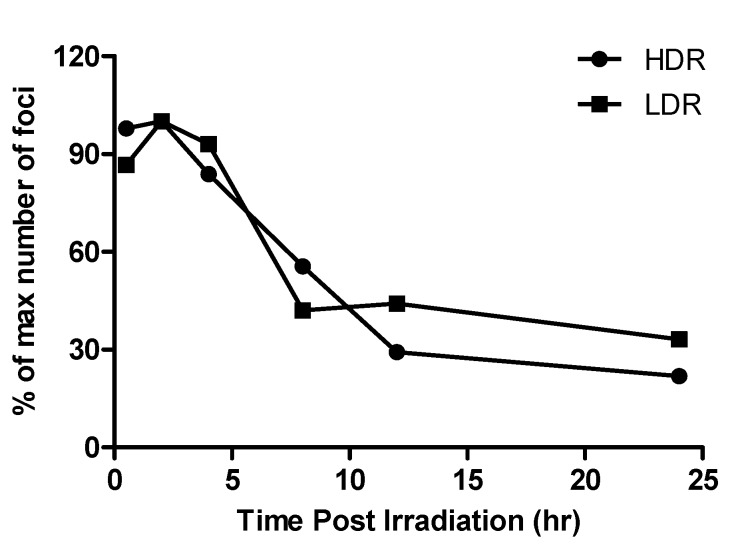
Postirradiation foci numbers were normalised to the maximum number of foci induced at 2 h postirradiation to 1 Gy of HDR (0.400 Gy/min) and LDR (0.015 Gy/min) neutrons. Data represents the mean number of γ-H2AX foci ± standard deviation of two independent experiments.

**Table 1 ijms-20-05350-t001:** Dose response data for γ-H2AX foci following exposure to high dose-rate (HDR; 0.400 Gy/min) and low dose-rate (LDR; 0.015 Gy/min) neutron radiation. All values represent the mean obtained for four different donors. The error bars are the standard deviations representing the interindividual variation among the four donors.

Dose Rate	Dose (Gy)				
Gy/min	0.125	0.250	0.500	1.000	2.000
**0.015**	1.25 ± 0.26	1.49 ± 0.36	2.85 ± 0.38	4.31 ± 0.44	5.45 ± 0.98
**0.400**	1.52 ± 0.17	2.79 ± 0.63	4.09 ± 0.19	5.61 ± 0.88	6.34 ± 0.91

**Table 2 ijms-20-05350-t002:** Ratio of the mean number of foci measured at different dose points (HDR/LDR).

Dose (Gy)				
0.125	0.250	0.500	1.000	2.000
1.22	1.87	1.44	1.30	1.16

**Table 3 ijms-20-05350-t003:** Repair kinetics for γ-H2AX foci following exposure to 1 Gy of HDR (0.400 Gy/min) and LDR (0.015 Gy/min) neutrons. All data represent the mean values obtained for four different donors.

Dose Rate	Time Postirradiation (h)					
Gy/min	0.5	2	4	8	12	24
**0.015**	4.32 ± 0.44	5.60 ± 0.48	4.61 ± 0.43	2.40 ± 0.31	2.21 ± 0.18	1.65 ± 0.46
**0.400**	5.63 ± 0.88	5.75 ± 0.05	4.89 ± 0.31	3.20 ± 0.63	1.68 ± 0.57	1.29 ± 0.45

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
