# Peer review of "The Impact of Dose Rate on DNA Double-Strand Break Formation and Repair in Human Lymphocytes Exposed to Fast Neutron Irradiation"

_ijms, 2019, doi:10.3390/ijms20215350_

Round 1

Reviewer 1 Report

Title: The impact of dose rate on DNA double strand break formation and repair in human lymphocytes exposed to fast neutron irradiation

Summary:

This manuscript describes the effects of fast neutron radiation on blood using DNA DSB as the endpoint.

Suggested revisions:

What is the impact of this type of irradiation on human health other than astronauts, does it have implications for radiotherapy for example? Statistics: were the data in Table 1 and Figure 2 significant? Was this analyzed? Is Figure 2 necessary, or can it be combined with Figure 1? Discussion: can the flow of information in the discussion be revised overall. It is very wordy and the message is often lost in this section of the manuscript. Rewriting of the discussion will help the reader understand the importance of this work.

Author Response

Suggested revisions:

What is the impact of this type of irradiation on human health other than astronauts, does it have implications for radiotherapy for example? Statistics: were the data in Table 1 and Figure 2 significant? Was this analyzed? Is Figure 2 necessary, or can it be combined with Figure 1? Discussion: can the flow of information in the discussion be revised overall. It is very wordy and the message is often lost in this section of the manuscript. Rewriting of the discussion will help the reader understand the importance of this work. 

Response: 

We thank the reviewer for the suggestions. The following modifications have been made to include the recommendations of the reviewer.

As the reviewer correctly indicates, neutrons are also a health concern in radiotherapy, particularly when high energy X-rays and protons are used to treat cancer patients. In radiotherapy with high-energy photon beams (energy > 10 MeV) neutrons are generated mainly in the head of the inac and the beam collimation system. During proton therapy, neutrons are inevitably produced through nuclear interactions in the components of proton beam line and in patients body. The stray radiation deposited outside the primary field may increase the risk of second malignancies. The latter is of particular importance for paediatric patients, known to be more radiosensitive and to have a longer life expectancy, which makes them especially susceptible to develop radiation-induced secondary cancer after radiotherapy. The secondary neutron doses are very low (typically <0.1% of the target dose) and the dose rate is not as low as applied in this study, but it is still a topic of debate if they can just be neglected.

The following sentence was added to the introduction part of the paper:

Introduction (line 114): Next to the health effects for astronauts during deep space missions, the long-term cancers risks related to secondary neutron production during radiotherapy remain a controversial topic, particularly when high energy X-rays (E > 10 MeV) and protons are used.

Discussion (line 201): In addition, a better understanding of the radiobiological effects of low neutron doses is relevant for radiotherapy patients where neutrons can form a significant part of the stray radiation produced during treatment.

The reviewer asked whether the data in Table 1 and Figure 2 were significant; it was concluded that the data was statically significant (p < 0.05). Both a paired and unpaired t-test was done on the complete group of data (on all dose points) to determine this result. Furthermore, Figure 2 was added to illustrate the sharp increase in the number of neutron-induced DNA DSBs at the lower doses and it appears to reach a plateau at 1 and 2 Gy. However, considering the suggested comment, we have decided to delete Figure 2, as the data is already presented in Table 1 and in Figure 1. We agree with the reviewers’ comment that Figure 1 and Figure 2 represent the same information and the reader can make this conclusion based on Figure 1.

Therefore, the manuscript has be adjusted such that Figure 2 has been deleted and the remaining figures have been renamed, as well as in the text.

We agree that the discussion is wordy, therefore, the authors removed some of the too extensive sentences and long descriptions, in order to create a better flow for the reader.

Reviewer 2 Report

This is a good and informative article adding much to the discussion of the dose rate dependence of biological effects of high LET radiation.

The following question might be posed in the manuscript: If there will be more such studies like that under review, maybe DDREF (or separately LDEF and DREF) values could be recommended not only for low LET but also for high LET radiation?

A few words should be added about a tendency to exaggerate harmful effects of low-dose low-rate exposures i.e. recommendation of exceedingly low DDREF values (down to 1). Motives of such exaggeration might be tackled e.g. strangulation of nuclear energy and boosting of fossil fuel prices (discussed in [1]). Radiation hormesis may be mentioned as well: within a certain range, the dose-effect relationship might become inverse due to hormesis, especially for low LET radiation [1].  

Jargin SV. Hormesis and radiation safety norms: Comments for an update. Hum Exp Toxicol. 2018;37(11):1233-1243. doi: 10.1177/0960327118765332

Author Response

Comments and Suggestions for Authors

This is a good and informative article adding much to the discussion of the dose rate dependence of biological effects of high LET radiation.

The following question might be posed in the manuscript: If there will be more such studies like that under review, maybe DDREF (or separately LDEF and DREF) values could be recommended not only for low LET but also for high LET radiation? A few words should be added about a tendency to exaggerate harmful effects of low-dose low-rate exposures i.e. recommendation of exceedingly low DDREF values (down to 1). Motives of such exaggeration might be tackled e.g. strangulation of nuclear energy and boosting of fossil fuel prices (discussed in [1]). Radiation hormesis may be mentioned as well: within a certain range, the dose-effect relationship might become inverse due to hormesis, especially for low LET radiation [1].  Jargin SV. Hormesis and radiation safety norms: Comments for an update. Hum Exp Toxicol. 2018;37(11):1233-1243. doi: 10.1177/0960327118765332 

Response: 

We thank the reviewer for the positive and encouraging remarks. The following modifications have been made to include the suggestions of the reviewer:

The suggested question on LDEF and DREF for high-LET radiation was added to the conclusion section of the manuscript, to make the statement stronger and to trigger the reader to take this into consideration after reading the manuscript.

Addition to the manuscript (line 304): These results are a first indication that the dose rate has an effect on the efficiency of DNA DSB induction and repair for fast neutrons, which contributes to the discussion whether a LDEF or DREF should be recommended for high-LET radiation.

We agree with the reviewer that the uncertainty on low dose risk estimations remains a challenge and might be exaggerated by regulatory bodies for various underlying reasons. Considering the comment of reviewer 1 on the length of the discussion, we tried to add this in the discussion of the paper as short but clear as possible. We decided to add the reference to the paper of Jargin for the readers who are interested to learn more about this topic, but didn’t describe the hormesis effect in details since the study was not designed to investigate this and it might add to the wordiness of the discussion.

For the estimation of stochastic effects such as solid cancers and leukaemia after external radiation exposure, most of our knowledge is based on the Life Span Studies (LSS) from the Japanese atomic bomb survivors, which were exposed to high-dose rates. In order to quantify radiation risks at exposure scenarios relevant for radiation protection, one has to extrapolate this data obtained at high doses and high-dose rates down to low-doses and low-dose rates [43]. However, this extrapolation comes with large uncertainties and the LNT theory remains a topic of active debate. Epidemiological studies on low dose effects remain challenging and a growing amount of evidence suggests that there is no scientific proof for low dose and low-dose rate effects. The current LNT approach might have a negative impact on public perception, policy statements and future directions for the radiation protection framework. Regulatory bodies have adopted the DDREF an approach for low-LET radiation which implies that radiation delivered at low total doses or low-dose rates are less effective for cancer induction, but not for high-LET radiation.